# Methane Admixture Protects Liver Mitochondria and Improves Graft Function after Static Cold Storage and Reperfusion

**DOI:** 10.3390/antiox12020271

**Published:** 2023-01-25

**Authors:** Tamara Horváth, Lilla Sándor, Bálint Baráth, Tibor Donka, Bence Baráth, Árpád Mohácsi, Kurszán Dávid Jász, Petra Hartmann, Mihály Boros

**Affiliations:** 1Institute of Surgical Research, University of Szeged, H-6724 Szeged, Hungary; 2Department of Traumatology, University of Szeged, H-6725 Szeged, Hungary; 3Department of Pathology, University of Szeged, H-6725 Szeged, Hungary; 4MTA–SZTE Research Group on Photoacoustic Spectroscopy, University of Szeged, H-6725 Szeged, Hungary

**Keywords:** liver transplantation, mitochondria, methane, reperfusion, static cold storage

## Abstract

Mitochondria are targets of cold ischemia-reperfusion (IR), the major cause of cell damage during static cold preservation of liver allografts. The bioactivity of methane (CH_4_) has recently been recognized in various hypoxic and IR conditions as having influence on many aspects of mitochondrial biology. We therefore hypothesized that cold storage of liver grafts in CH_4_-enriched preservation solution can provide an increased defence against organ dysfunction in a preclinical rat model of liver transplantation. Livers were preserved for 24 h in cold histidine–tryptophan–ketoglutarate (HTK) or CH_4_-enriched HTK solution (HTK-CH_4_) (n = 24 each); then, viability parameters were monitored for 60 min during normothermic isolated reperfusion and perfusate and liver tissue were collected. The oxidative phosphorylation capacity and extramitochondrial Ca^2+^ movement were measured by high resolution respirometry. Oxygen and glucose consumption increased significantly while hepatocellular damage was decreased in the HTK-CH_4_ grafts compared to the HTK group. Mitochondrial oxidative phosphorylation capacity was more preserved (128.8 ± 31.5 pmol/s/mL vs 201.3 ± 54.8 pmol/s/mL) and a significantly higher Ca^2+^ flux was detected in HTK-CH_4_ storage (2.9 ± 0.1 mV/s) compared to HTK (2.3 ± 0.09 mV/s). These results demonstrate the direct effect of CH_4_ on hepatic mitochondrial function and extramitochondrial Ca^2+^ fluxes, which may have contributed to improved graft functions and a preserved histomorphology after cold IR.

## 1. Introduction

Liver transplantation is the ultimate therapeutic option in end-stage liver diseases and the clinical success of the intervention is dependent on the effectiveness of allograft protection from the time of donation to surgical transplantation [1]. Several liver transplantation concepts have been attempted to date, but static cold storage (SCS) in a special preservation solution is the most accredited organ preservation modality [2]. However, ischemic graft injury is still inevitable due to anaerobic metabolism, which leads to a depletion of energy stores and concomitant loss of cellular function and integrity. It follows that mitochondria are target intracellular organelles of ischemia-induced changes. The lack of the final electron acceptor (oxygen) collapses the electron transport chain, resulting in an accumulation of respiratory complex substrates (NADH and succinate), while ATP generation through oxidative phosphorylation ceases [3,4,5,6,7]. The depletion of ATP leads to the dysfunction of membrane sodium–potassium pumps (Na^+^/K^+^ ATPases), the re-uptake of Ca^2+^ ions via calcium pumps (Ca^2+^ ATPases) on the endoplasmic reticulum (ER) and intracellular enzymatic reactions [8,9]. Upon reperfusion, further mitochondrial damage supervenes when the accumulated succinate is rapidly oxidized, driving superoxide production at complex I by reverse electron transport, resulting in oxidative damage to the mitochondria, opening the mitochondrial permeability transition pores (mPTP) and eventually leading to cell death [3,4,10]. It has been recognized that mitochondrial metabolic changes significantly contribute to poor post-transplant graft functions [8,11]. Mitochondrial protection has therefore come into focus for the development of cold storage organ preservation solutions [12,13,14]. 

In recent years, several preclinical studies have demonstrated improved donor organ quality after administration of biologically active gases. This approach is mainly based on the notion that gas molecules require no special receptors or active transport to diffuse into the internal milieu of the cells [8,15,16]. One such gas, methane (CH_4_), can reach a high concentration when dissolved in water or colloid solutions, such as a preservation solution, and it has been proven to be effective in various hypoxic and inflammatory scenarios [8]. CH_4_ is the most reduced form of carbon and part of the omnipresent gaseous environment which maintains aerobic metabolism within the eukaryote cell. It is intrinsically non-toxic and a simple asphyxiant, meaning that tissue hypoxia might occur when the gas displaces air in a restricted area. The first reports of the protective effect of CH_4_ against oxygen-mediated and reactive oxygen species (ROS)-mediated oxidoreductive stress [17] were followed by a number of studies where the anti-inflammatory and anti-apoptotic properties of CH_4_-based treatments were demonstrated in liver and cardiac ischemia-reperfusion (IR) and IR-associated animal models [3,4,8,18,19]. These studies have already confirmed that many aspects of mitochondrial biology are involved in the mechanism of CH_4_ actions. In a partial liver IR model, administration of normoxic CH_4_ during the reperfusion phase effectively restored the electron transport chain capacity and reduced the cytochrome c release and ROS formation of hepatic mitochondria [3]. In a cardiac transplantation model, CH_4_ supplementation of the grafts preserved the oxidative phosphorylation capacity of mitochondria and significantly reduced the relative mRNA expression for hypoxia- and ER–mitochondria stress-associated genes (including HIF-1α) in the rat heart [8]. 

Considering such properties, exogenous administration of CH_4_ seems to be an attractive strategy to achieve mitochondrial protection during cold storage of liver grafts. To date, the effects of CH_4_ in terms of liver transplantation have not yet been investigated, and, therefore, the main purpose of our study was to test the suitability and effectiveness of a modified, CH_4_-enriched preservation solution on post-transplant liver function and integrity. We planned to employ a preclinical liver transplantation model where livers from cardiac-dead donor rats are exposed to CH_4_-enriched histidine–tryptophan–ketoglutarate (HTK) solution during 24 h of cold ischemia, then oxygenated and re-warmed in an ex vivo system. In this model, we used simple tests to judge the therapeutic potential of CH_4_ on active organ metabolism, including oxygen and glucose consumption and lactate and bile production. The mechanism underlying CH_4_ action was studied further and metabolic changes were assessed on-site at the level of mitochondria with high resolution respirometry. 

## 2. Materials and Methods

The experiments were carried out on male Sprague–Dawley rats (265–325 g; Charles River, Budapest, Hungary) in accordance with EU Directive 2010/63 for the protection of animals used for scientific purposes and in compliance with criteria laid out in the US National Institutes of Health Guidelines for the Care and Use of Laboratory Animals. The study was approved by the national competent authority of Hungary (ATET) under license number V./1416/2021.

### 2.1. Production of CH_4_-Enriched HTK

Commercially available Custodiol solution (Dr. Franz Köhler Chemie GmbH, Bensheim, Germany) was saturated with pure CH_4_ (>99.9%) under 0.4 MPa for 4 h in a high-pressure vessel (Messer, Budapest, Hungary) to enrich HTK with CH_4_, as described previously [18]. The CH_4_ concentration in the fluid phase was detected by gas chromatography, while the stability of the solution was checked by near-infrared laser-based photoacoustic spectroscopy, as described earlier [4]. The solution containing 6.57 ± 0.27 μmol/mL CH_4_ was freshly prepared and stored at 4 °C before use.

### 2.2. Experimental Groups

The animals were anesthetized with ketamine (45.7 mg/kg i.p.) and xylazine (9.12 mg/kg i.p.) and randomly allocated into three groups. The liver retrieval procedure was performed, as described previously [20]. The portal vein, the superior branch of the inferior vena cava and the common bile duct were then cannulated under an operating microscope. The grafts were perfused ex vivo with an organ perfusion system (Central European Biosystem, FALC Instruments, Budapest, Hungary) through the afferent branch of the portal vein, while the caval vein served as the efferent part of the system. The warm ischemic group (WI, *n* = 24) was the control group where the grafts were perfused immediately after retrieval without being subjected to cold ischemia. In the HTK (*n* = 24) and CH_4_-enriched HTK (HTK + CH_4_, *n* = 24) groups, the grafts were stored in HTK solution at 4 °C for 24 h (Figure 1).

### 2.3. Organ Perfusion System and Assessment of Hepatic Function and Cell Injury

Reperfusion was performed using an isolated rat liver perfusion system (IPRL), as described previously in detail [21]. Grafts were perfused with 300 mL of oxygenated, modified Krebs–Henseleit buffer at 37 °C for 60 min at a constant flow of 3 mL/g of liver weight/minute using a roller pump (Masterflex, L/S, Cole Parmer Instrument Company, Vernon Hills, IL., USA). Portal venous pressure (PVP) was continuously monitored via the water column attached to the afferent side of the perfusion system and bile was collected in an Eppendorf tube through a cannula attached to the common bile duct. The hepatic perfusate samples (2 mL) were collected intermittently from both sides of the system for pH-fluid gas analysis (Cobas b123; Roche Ltd., Basel, Switzerland) and the oxygen and glucose uptakes were determined as the difference between portal (inflow) and central venous (outflow) concentrations, expressed as microliters per gram of liver weight per minute. Spectrophotometric measurements of liver necroenzymes, such as alanine aminotransferase (ALT), aspartate aminotransferase (AST) and lactate dehydrogenase (LDH), were measured from the outflow perfusate samples [22]. 

### 2.4. High Resolution Respirometry

Mitochondrial respiration was measured with Oxygraph-2k (Oroboros Instruments, Innsbruck, Austria). Liver samples were taken at the end of reperfusion and homogenates were prepared in a mitochondrial respiration medium (MiR05). 100 µL of the suspension was then added to an experimental chamber of 2 mL volume and oxygen consumption of different mitochondrial metabolic states was measured. The baseline respiration of liver cells was defined without substrates and inhibitors. Next, we determined individual complex I and complex II respiratory activities by adding 10 mM glutamate and 2 mM malate simultaneously and in the next step, 10 mM of succinate was added to stimulate complex I and complex II, respectively. Complex V substrate of 5 mM ADP was added to measure oxidative phosphorylation in state 3 respiration. Subsequently, we added 0.5 µM of ATP synthase inhibitor oligomycin to the chambers to induce ATP-independent respiration (LEAK) and provide a biochemical gradient for further steps. Oxidative phosphorylation and LEAK respiration data were used as a respiratory control ratio, which is directly but non-linearly related to the oxidative phosphorylation coupling efficiency and expressed as the oxidative phosphorylation/LEAK ratio. Furthermore, inner membrane injury was determined by adding 5 mM of cytochrome c, a heme protein loosely associated with the outer side of the inner mitochondrial membrane. Finally, 1 μM of complex III inhibitor antimycin A was administered to completely block oxidative phosphorylation, thus providing information on background respiratory activity. 

The mitochondrial Ca^2+^ flux was measured using the blue fluorescence module (excitation: 465 nm; gain: 1000; polarization voltage: 500 mV). Ca^2+^ flux evoked by external use of CaCl_2_ was ascertained using CaGreen-5N monofluorescent dye. This method is suited to the evaluation of Ca^2+^ fluxes even in high concentrations due to its low affinity for Ca^2+^ (K_D_ 14,000 nM, 0.5–50 μM) [23]. In the first step, the baseline Ca^2+^ flux was determined in the absence of any triggering of Ca^2+^ release with the addition of fluorescent dye (2 µL of CaGreen-5N). This Ca^2+^ flux refers to the baseline level of extramitochondrial Ca^2+^ from endogenous Ca^2+^ sources. Then, the oxidative phosphorylation was reached with the addition of succinate and ADP substrates to the sample, while an inhibitor of complex I rotenone (1 μM) was added to initiate mitochondrial Ca^2+^ fluxes. This was followed by a 3-step titration of exogenous CaCl_2_ (5 mM) to reach the maximum Ca^2+^ flux, which indicates mPTP opening. After the supersaturating effect of CaCl_2_, electron transport was blocked with complex III inhibitor antimycin A, inhibiting mitochondrial Ca^2+^ fluxes while the system Ca^2+^ potential was stopped with ethylene glycol tetra-acetic acid (1 mM EGTA). 

### 2.5. Histology and Immunohistochemistry

Tissue specimens were fixed in buffered paraformaldehyde solution (4%), embedded in paraffin and 5-μm-thick sections were cut and then stained with hematoxylin and eosin (H&E). The morphology of liver integrity was graded according to criteria developed for IPRL livers [24] on a scale of 1 (excellent) to 5 (poor) described as: (1) normal rectangular structure, (2) rounded hepatocytes with increased sinusoidal spaces, (3) vacuolization, (4) nuclear pyknosis and (5) necrosis.

Endothelial cell injury was classified by the degree of cell detachment from the vascular matrix, discriminating between arterioles, large veins and small venules. Two operators independently but concurrently examined at least 15 microscopic fields of three different biopsies by light microscopy. 

The ER stress marker SERCA2b (Endoplasmic Reticulum Ca^2+^ ATPase) (#PA5-102354, Thermo Fisher Scientific Inc., Waltham, MA, USA) was detected by immunohistochemical staining, as previously described [4]. 

### 2.6. Statistics

Data analysis was performed with Sigma Plot 13.0 statistical software (Jandel Corporation, San Rafael, CA, USA). Changes in variables within and between groups were analyzed by two-way repeated measures ANOVA, followed by Tukey’s test and the Kruskal–Wallis test. All data were expressed as means ± SD. Values of *p* < 0.05 were considered statistically significant.

## 3. Results

### 3.1. Graft Function and Cell Injury after Cold Storage

Cold storage in HTK resulted in a significant drop in hepatic oxygen uptake throughout the reperfusion period compared to the WI group. The HTK + CH_4_ group showed significantly better oxygen uptake during the whole reperfusion period, peaking at 277.28 ± 30.04 µL/glw/min at 45 min vs. 183.89 ± 18.8 µL/glw/min in the HTK group, but still remaining lower than the WI group (381.92 ± 30.32 µL/glw/min) (Figure 2A).

In the HTK group, cold storage significantly impaired glucose uptake from minutes 5 to 45 compared to the WI group (from 0.14 ± 0.03 to 0.15 ± 0.02 µmol/glw/min vs 0.33 ± 0.01 to 0.28 ± 0.01 µmol/glw/min). These results in the HTK + CH_4_ group were significantly higher than those in the HTK group at 30 and 45 min of reperfusion (0.20 ± 0.01 and 0.22 ± 0.02 µmol/glw/min) (Figure 2B).

In the HTK group, a steady rise in lactate level was observed during reperfusion, reaching 1.3 ± 0.18 mmol/l at minute 45. CH_4_ treatment significantly attenuated the increase in the HTK + CH_4_ group, which was 0.84 ± 0.08 mmol/l at minute 45. Lactate levels were significantly lower in the WI group than the cold-stored groups and did not change from minutes 5 to 45 during the reperfusion period (from 0.39 ± 0.07 to 0.40 ± 0.07 mmol/L) (Figure 2A). 

The total vascular resistance of the liver graft during reperfusion was measured as PVP. The WI group had lower levels of PVP (9.89 ± 2.7 to 9.03 ± 1.7 mm Hg) in comparison to the other groups throughout the reperfusion period. The HTK + CH_4_ group showed significantly lower values (19.86 ± 2.2 to 16.9 ± 1.3 mm Hg) compared to the HTK (24.50 ± 2.7 to 22.9 ± 3.1 mm Hg) group during reperfusion (Figure 3A).

Bile was collected throughout the reperfusion period. Bile production was higher in the WI group (36.7 ± 4.9 μL/45 min) as compared to the HTK group (23.2 ± 3.6 μL/45 min) and the HTK + CH_4_ (31.3 ± 3.5 μL/45 min) groups; nevertheless, there was a significant difference between the cold-stored groups, and CH_4_ supplementation provided better bile excretion in the HTK + CH_4_ group (Figure 3B).

The release of ALT after 45 min of reperfusion was significantly higher for the HTK group (90.0 ± 19.4 U/L) versus the WI group (9.1 ± 1.9 U/L; Figure 4A). Compared to the HTK group, the HTK + CH_4_ group (51.7 ± 15.7 U/L) demonstrated statistically significantly lower levels (*p* < 0.05). The AST and LDH levels were statistically lower in the WI group than the cold-stored groups. The HTK + CH_4_ group tended to show lower AST and LDH values throughout the reperfusion period (Figure 4B,C). However, only LDH displayed statistically significant elevation in the HTK group after 30 and 45 min of reperfusion (2100 ± 427 U/L at 45 min) compared to the HTK + CH_4_ group (1541 ± 202 U/L) (Figure 4C). 

### 3.2. Mitochondrial Respiration

Changes in mitochondrial respiratory functions were evaluated in the presence of glutamate and malate or succinate to differentiate between complex I- and complex II-based activity. Cold-stored groups displayed significantly reduced complex I activity (19.6 ± 11.7 pmol/s/mL in HTK and 22.0 ± 10.9 pmol/s/mL in HTK + CH_4_) and reduced oxidative phosphorylation (128.8 ± 31.5 pmol/s/mL in HTK and 201.3 ± 54.8 pmol/s/mL in HTK + CH_4_) in the presence of a saturating amount of ADP as compared to the control group (WI; 26.7 ± 7.6 pmol/s/mL and 280.2 ± 64.5 pmol/s/mL, respectively). The decrease in complex I activity and the related oxidative phosphorylation was significantly higher in the hepatic mitochondria of the HTK grafts than the HTK + CH_4_-preserved grafts. However, in the presence of the complex II substrate succinate, hepatic mitochondria displayed similar respiratory activity in all the study groups (65.9 ± 16.2 pmol/s/mL in control, 55.5 ± 24.6 pmol/s/mL in HTK and 59.8 ± 7.6 pmol/s/mL in HTK + CH_4_). The respiratory acceptor control ratio (RCR), expressed as the oxidative phosphorylation/oligomycinLEAK ratio, was calculated to quantify changes in the coupling of the electron transport chain [25]. Compared to the WI group, the respiratory control ratio indicated a significant impairment of the electron transport chain in the HTK group (1.4 ± 0.2). However, no significant difference was found between the HTK + CH_4_ and WI groups in the coupling of mitochondria (2.7 ± 0.6 and 4.9 ± 0.6, respectively) (Figure 5).

### 3.3. Extramitochondrial Ca^2+^ Movement 

The cold ischemia-related endogenous Ca^2+^ release was assessed at first as the baseline Ca^2+^ flux of the samples. There was a significant increase in the CaGreen-5N fluorescent intensity in the HTK group (1.9 ± 0.1 mV/s) as compared to the control group (WI; 1.3 ± 0.04 mV/s and 1.4 ± 0.1 mV/s, respectively). Adding CH_4_ admixture to the preservative solution in the HTK + CH_4_ group resulted in a significant decrease in fluorescent intensity (Figure 6B). The exogenous Ca^2+^-triggered Ca^2+^ flux was then examined by adding a saturating amount of CaCl_2_, thus inducing the mPTP opening. As a result of prolonged cold ischemia, mitochondria in the HTK group exhibited less responsiveness of fluorophore to exogenous Ca^2+^ (2.3 ± 0.09 mV/s) than hepatic mitochondria in the WI group (3.8 ± 0.2 mV/s). A significantly higher Ca^2+^ flux was detected in the HTK + CH_4_ group (2.9 ± 0.1 mV/s) as a result of CH_4_ supplementation (Figure 6C). 

### 3.4. Histology 

Liver morphology revealed viable hepatocytes in the control group (WI). For the HTK group, the hepatocellular damage was more pronounced, resulting in a significantly higher histological grade. For the HTK + CH_4_ group, the grade of injury was also elevated, although to a lesser extent than in the HTK group, thus indicating increased tissue protection. Endothelial injury was found in all the groups, particularly in the large vessels. Of note, the HTK group also showed endothelial detachment in the small vessels (Figure 7A–D). The number of SERCA2b immunoreactive hepatocytes decreased significantly in sections from HTK-stored grafts as compared to the controls (the WI group) (Figure 6E,F). In contrast, the number of immunoreactive cells was significantly increased in the HTK + CH_4_ group (Figure 7E–H).

## 4. Discussion

In the present study, we investigated the consequences of CH_4_ enrichment of HTK solution during cold static storage in a preclinical model of liver transplantation. CH_4_ supplementation resulted in superior morphology and functional hepatoprotection during a standardized IR injury. Markedly improved metabolic parameters, such as oxygen and glucose consumption, decreased PVP and necroenzyme levels and increased bile production was demonstrated. Furthermore, CH_4_ supplementation during cold storage significantly influenced the critical components of the respiratory mechanism of hepatic mitochondria. All in all, these results comprehensively demonstrate the direct role the CH_4_ supplementation of preservation solution performs in preserving the metabolic activity of liver cells. Additionally, these results have identified a form of mitochondrial protection that allows for better graft functions. The metabolic homeostasis of the liver depends on the supply of oxygen; the function of the mitochondria is thus severely affected by prolonged ischemia [26]. Since hepatic IR injury is the main cause of early graft loss and poor post-transplant function in liver transplantation, finding an appropriate preservation solution is an important clinical goal [27,28]. HTK solution has been adopted in the routine clinical practice of liver transplantation with equivalent results as an alternative to the gold standard University of Wisconsin (UW) solution [29,30,31,32]. Of the preservative solutions, HTK is the most effective in preventing biliary complications, with its low viscosity providing a better initial flush of the liver with more rapid cooling; additionally, it can be successfully enriched with CH_4_ [8,29]. CH_4_ is an amphipathic compound, which is a property that enables quick and easy transport through biological membranes without requiring specific transporters or receptors [33]. Therefore, CH_4_ can reach a higher concentration in lipid bilayers, such as the mitochondrial membrane, and ROS generation can lead to a higher level of fixation [34]. Once considered an inert gas, CH_4_ was proven to affect multiple biochemical processes, resulting in changes on the cellular and subcellular levels and thus leading to the modulation of hypoxic damage and inflammatory responses. Notably, it has been shown that exogenous CH_4_ modulates the intrinsic, mitochondrial pathway of pro-apoptotic activation in model experiments [35,36] and preserves the mitochondrial respiratory capacity in cells exposed to anoxia [4]. Furthermore, CH_4_ significantly reduced the expression of hypoxia- and ER stress-associated genes in experimental heart transplantation [8]. A prerequisite for successful liver transplantation is immediate graft function, as there is currently no suitable method to supplement the metabolism of the liver. In the liver transplantation model presented, the ability of the liver to extract oxygen and glucose from the perfusate fluid and produce lactate served as functional tests to assess the activity of organ metabolism. The hypothermic preservation in HTK solution resulted in the reduced oxygen and glucose uptake of liver grafts, while hepatocellular damage, confirmed by transaminase release, was increased, a finding which also demonstrates that currently used preservation methods cannot achieve complete tissue protection against cold IR-induced damage. The LDH, AST and ALT content of the outflow perfusate fluid indicated general hepatic injury, all referring to the extent of hepatocellular damage, but ALT is more specific to liver sinusoidal injury [37]. The liver grafts in the HTK + CH_4_ group demonstrated much lower levels of all necroenzymes during the reperfusion period as compared to the HTK grafts, a result which points to the overall protective effect of CH_4_ on organ viability.

In human liver transplantation, high post-transplant PVP is associated with higher incidences of primary dysfunction and non-function of the graft, probably due to the impaired oxygen supply to the hepatocytes through the sinusoidal endothelial cells [38]. In our model, PVP was significantly reduced in the HTK + CH_4_-treated liver compared to the HTK grafts. The lower PVP of the HTK + CH_4_-treated livers could have been related to the local, tissue-specific effect of CH_4_ on endothelial cells, which may have prevented vascular occlusion [39]. Alternatively, CH_4_ has been demonstrated to influence ROS-generating xanthine oxidoreductase activity and nitrogen biology in the hypoxic tissue, suggesting an interplay between the vasodilator nitrogen monoxide and CH_4_ in the intracellular milieu [40]. The liver has very high energy requirements for the active secretion of bile, a process that is sensitive to IR damage [20,41]. Bile production is therefore an indicator of the viability of isolated perfused liver grafts [42,43]. The HTK + CH_4_-treated grafts demonstrated an increased secretion of bile compared to the HTK group, a result which could be linked to the effects of CH_4_ on preserved oxidative phosphorylation in mitochondria, which enables energy-demanding bile secretion.

Liver parenchymal and epithelial cells, hepatocytes and cholangiocytes contain 1000–2000 mitochondria per cell to sustain metabolic activity. As a result of CH_4_ enrichment of the preservative solution, graft mitochondria were more responsive to ADP utilization, thus contributing to the maintenance of oxidative phosphorylation capacity. The individual capacities of the respiratory complexes were also tested with specific substrates and inhibitors. Interestingly, complex I activity was strongly diminished in the cold-stored HTK group, while there was no significant difference in complex II activity between the groups. This result highlights the central role of complex I in mitochondrial dysfunction during cold storage, setting it as the main target of future graft preservation therapies [2,44,45]. Furthermore, it strengthens previous findings on the mechanism of CH_4_ action, which have demonstrated a partial blockade of electron transport in complex I as a result of CH_4_ administration in a primary mitochondrial model of cardiac IR injury [4]. Specifically, the CH_4_ treatment of mitochondria restricted the forward electron transfer within complex I under normoxic conditions while effectively restricting reverse electron transport in post—anoxic mitochondria [4]. In the present study, in addition to the interaction with complex I activity, a non-specific action of CH_4_ was demonstrated by an increased oxidative phosphorylation capacity, decreased cytochrome c release and ameliorated LEAK respiration in the HTK + CH_4_ group, which was linked to the preserved membrane integrity and electron transfer capacity of mitochondria.

Previous studies have already suggested that the mechanism of action of CH_4_ supplementation influences mitochondrial physiology and Ca^2+^ homeostasis [8,18]. Depletion of mitochondrial substrates during ischemia is a major contributor to the development of intracellular hypercalcemia-related cellular damage [9,10]. The contribution of ER through the activation of ER-mediated Ca^2+^ transport in the mechanism of cold ischemia-induced cellular damage is relatively well characterized [46]. Hypoxia induces Ca^2+^ release from the ER and the re-uptake into the ER is regulated by SERCAs pumps, of which the SERCA3 isoform was found to decrease in the liver after IR injury [47]. In this experiment, cold storage caused reduced SERCA expression, which was attenuated by CH_4_ supplementation. The resulting elevated cytosolic Ca^2+^ levels cause raised Ca^2+^ uptake (influx) into the mitochondria and mitochondrial Ca^2+^ overload, in turn leading to mPTP opening, mitochondrial depolarization and the initiation of cell death [48,49]. The mechanism behind the Ca^2+^ influx-mediated membrane dysfunction may involve the mitochondrial Na^+^/Ca^2+^ exchanger, the H^+^/Ca^2+^ exchanger and mPTPs and their combined action [50,51]. In our model experiment, the 24-h-long ischemic insult induced a significant rise in the level of extramitochondrial Ca^2+^ from endogenous Ca^2+^ sources. CH_4_ supplementation of the preservation solution effectively attenuated this Ca^2+^ release by enhancing re-uptake into the ER, restoring mitochondrial membrane integrity and, indirectly, preserving the function of energy-consuming membrane-associated Ca^2+^ ion channels. The function of mPTPs was also evaluated in this study, when hepatic mitochondria were subjected to exogenous Ca^2+^ overload, which is a trigger for mPTP opening. As a consequence of the cold IR, the mitochondrial response to external Ca^2+^ loading was associated with a low plateau of Ca^2+^ efflux in the HTK group, suggesting early or existing mPTP opening. Preservation of the grafts in CH_4_-supplemented transplantation solution prevented ischemia-induced mPTP opening, as indicated by higher Ca^2+^ efflux in the HTK + CH_4_ group. 

## 5. Conclusions

In conclusion, our results demonstrate that the preservation of donor livers in a CH_4_-supplemented solution is an effective pharmacological approach to reduce cold ischemia-induced damage and limit hepatic complications. Experimental evidence demonstrates that CH_4_ can alleviate interrelated intracellular events involving mitochondrial respiration and extramitochondrial Ca^2+^ movements, leading to preserved hepatocellular integrity and liver function. Based on these experimental results, it seems reasonable to propose that the addition of CH_4_ admixture to preservation solutions will provide more effective means for clinical organ transplantation protocols as well.

## Figures and Tables

**Figure 1 antioxidants-12-00271-f001:**
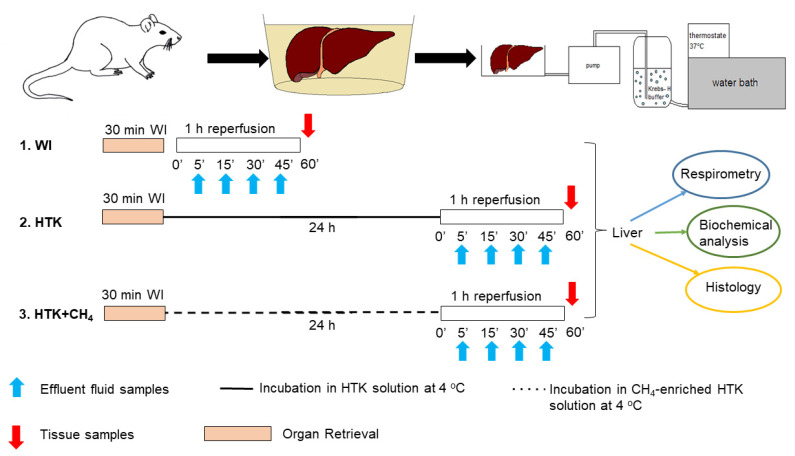
Study protocol. Preparation and perfusion of donor livers. Grafts in the WI group were subjected to immediate perfusion following retrieval. The other groups were incubated in HTK and CH_4_-enriched HTK solution (HTK + CH_4_), respectively, at 4 °C for 24 h prior to organ perfusion. Perfusate samples were obtained during reperfusion for biochemical analysis (indicated with blue arrows). At the end, tissue samples were taken for further analysis (indicated with red arrows). WI: warm ischemia; HTK: histidine–tryptophan–ketoglutarate; CH_4_: methane; HTK + CH_4_: CH_4_-enriched HTK.

**Figure 2 antioxidants-12-00271-f002:**
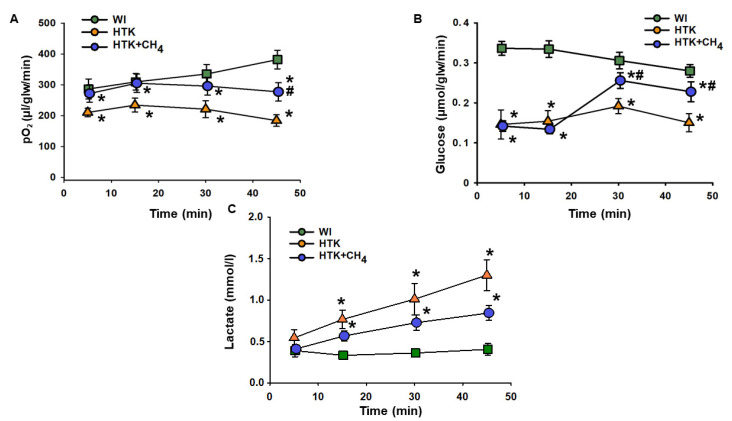
Liver metabolic parameters. (**A**) Oxygen consumption determined as the difference in partial oxygen pressure of influent and effluent Krebs solution. (**B**) Glucose consumption determined as the difference in glucose concentration of influent and effluent Krebs solution. (**C**) Lactate content measured from effluent Krebs solution. The WI group is marked with a green square, the HTK group is marked with a yellow triangle and the HTK + CH_4_ group is marked with a blue circle. Data are presented as means ± SD. * *p* < 0.05 vs. WI; # *p* < 0.05 vs. HTK (one-way ANOVA and Tukey’s test).

**Figure 3 antioxidants-12-00271-f003:**
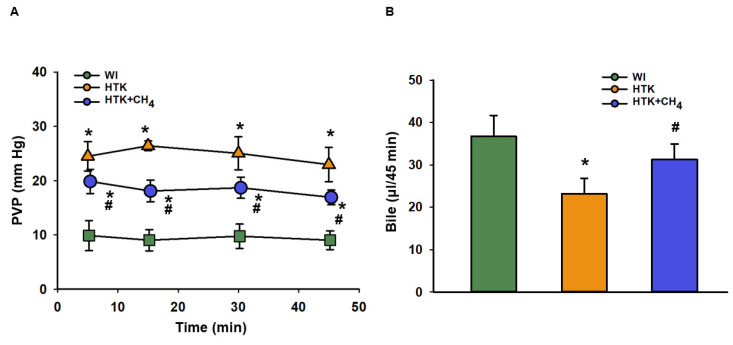
Portal venous pressure and bile production of liver grafts (**A**) Changes in pressure in the portal venous system during the perfusion period. (**B**) Cumulative bile production during perfusion. The WI group is marked green, the HTK group is marked yellow and the HTK + CH_4_ group is marked blue. Data are presented as means ± SD. * *p* < 0.05 vs. WI; # *p* < 0.05 vs. HTK (one-way ANOVA and Tukey’s test).

**Figure 4 antioxidants-12-00271-f004:**
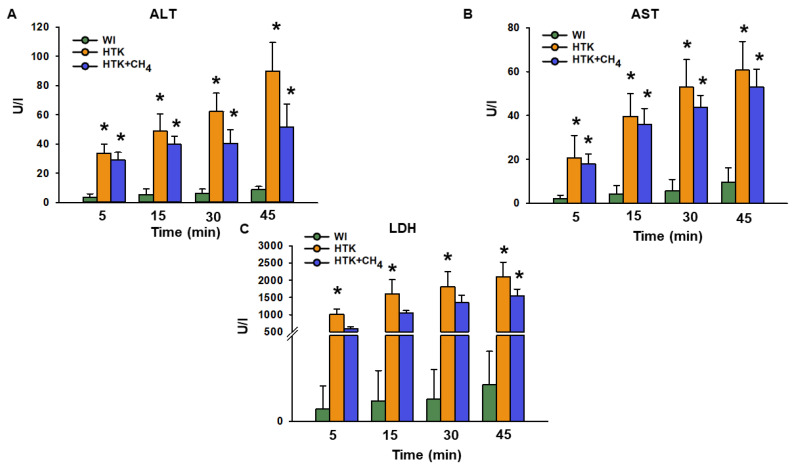
Liver necroenzymes. (**A**) Alanine aminotransferase (ALT), (**B**) aspartate aminotransferase (AST) and (**C**) lactate dehydrogenase (LDH) measured from effluent Krebs solution. Data at 45 min represent the cumulative necroenzyme content. The WI group is marked green, the HTK group is marked yellow and the HTK + CH_4_ group is marked blue. Data are presented as means ± SD. * *p* < 0.05 vs. WI; (one-way ANOVA and Tukey’s test).

**Figure 5 antioxidants-12-00271-f005:**
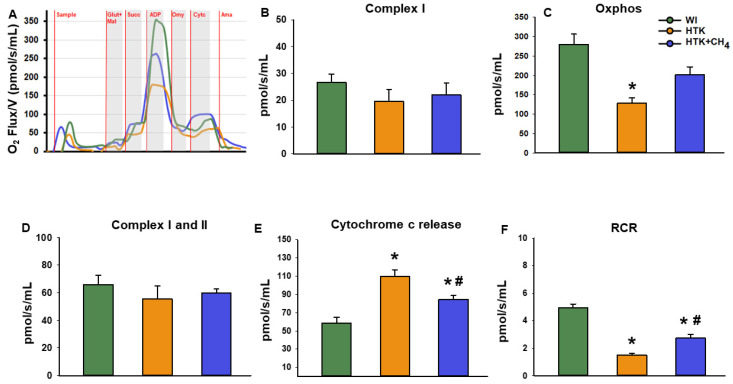
Mitochondrial respiration. (**A**) Representative records of mitochondrial oxygen consumption measured by high resolution respirometry. (**B**) Complex I-linked (NADH-generating substrates: glutamate and malate-dependent) respiration, (**C**) Oxidative phosphorylation (ADP-dependent respiration) in the presence of saturating levels of substrates. (**D**) Complex I and II-linked (added succinate respiration). (**E**) Cytochrome c release assessed by adding exogenous cytochrome c to the sample in the presence of glutamate and malate or succinate. (**F**) Respiratory control ratio (RCR) expressed as a ratio of ADP-dependent oxidative phosphorylation and ADP-independent LEAK respiration. The WI group is marked green, the HTK group is marked yellow and the HTK + CH_4_ group is marked blue. Data are presented as means ± SD. * *p* < 0.05 vs. WI; # *p* < 0.05 vs. HTK (one-way ANOVA and Tukey’s test). Glutamate (Glut), malate (Mal), Oxidative phosphorylation (Oxphos), succinate (Succ), cytochrome c (Cytc), antimycin A (Ama).

**Figure 6 antioxidants-12-00271-f006:**
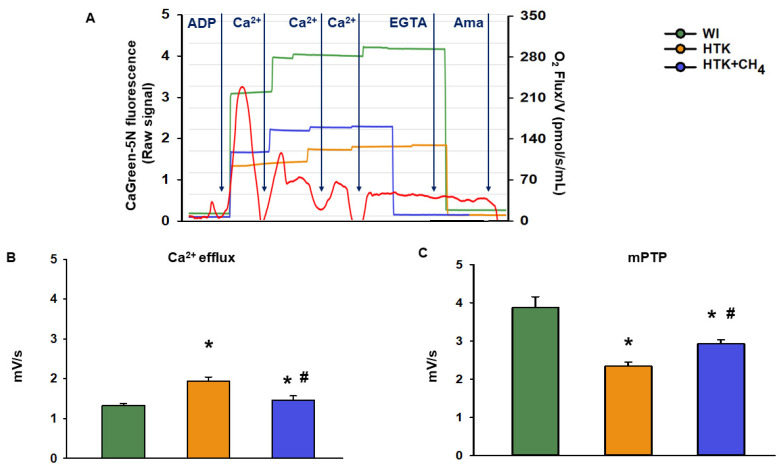
Mitochondrial Ca^2+^ fluxes. (**A**) Representative records of extramitochondrial Ca^2+^ fluxes in the WI (green line), HTK (yellow line) and HTK + CH_4_ groups (blue line) superimposed on the mitochondrial oxygen consumption (red line). (**B**) Endogenous Ca^2+^ release measured during cold ischemia in state 3 respiration. (**C**) Exogenous Ca^2+^-triggered maximum Ca^2+^ flux following CaCl_2_ titration, which indicates mPTP opening. The WI group is marked green, the HTK group is marked yellow and the HTK + CH_4_ group is marked blue. Data are presented as means ± SD. * *p* < 0.05 vs. WI; # *p* < 0.05 vs. HTK (one-way ANOVA and Tukey’s test).

**Figure 7 antioxidants-12-00271-f007:**
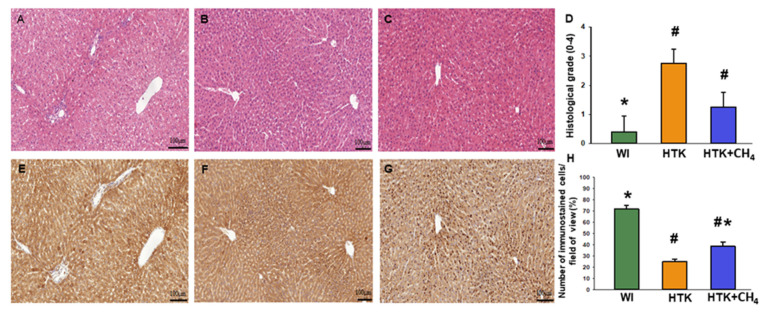
Histology and immunohistochemistry. (**A**,**E**) The WI group, (**B**,**F**) the HTK group and (**C**,**G**) the HTK-CH_4_ group. The bar represents 100 μm. (**D**) Histological grading of groups represents a composite of number of damaged hepatocytes (*n* = 15–18). Histological sections were stained with hematoxylin and eosin. Morphological liver integrity was graded according to criteria developed for IPRL livers on a scale of 1 (excellent) to 5 (poor) described as follows: (1) normal rectangular structure, (2) rounded hepatocytes with increased sinusoidal spaces, (3) vacuolization, (4) nuclear pyknosis and (5) necrosis. Endothelial cell injury was classified by the degree of cell detachment from the vascular matrix discriminating between arterioles, large veins and small venules (**H**). Immunostained cells in the field of view plotted as a percentage. The WI group is marked green, the HTK group is marked yellow, and the HTK + CH_4_ group is marked blue. Data are presented as means ± SD. * *p* < 0.05 vs. WI; # *p* < 0.05. vs. HTK.

## Data Availability

All the data are available within the article.

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
