# Peer review of "Methane Admixture Protects Liver Mitochondria and Improves Graft Function after Static Cold Storage and Reperfusion"

_antioxidants, 2023, doi:10.3390/antiox12020271_

Round 1
Reviewer 1 Report
The manuscript by Tamara Horváth and coauthors described the rat liver graft function after static cold storage and reperfusion. Livers were preserved for 24 h in cold histidine-tryptophan-ketoglutarate (HTK) or CH4-enriched HTK (HTK-CH4) solutions. After that viability parameters were monitored for 60 min during normothermic isolated reperfusion. CH4 supplementation of solution resulted in superior morphology and functional hepatoprotection.
The authors concluded that preservation of donor livers in CH4-supplemented solution is an effective pharmacological approach to reduce cold ischemia-induced damage and limit hepatic complications.
I think that the article will be interesting for the readers of the Journal.
My comments:
1. There are the statistically significant differences between HTK and HTK+CH4 groups during the whole period of reperfusion according to Fig. 2a.
But in the text “at minute 45 HTK+CH4 group showed significantly better oxygen uptake than the HTK group”.
2. Did the authors measure the effect of CH4 enrichment of solution on the oxygen concentration in the preservation solution?
Reviewer 2 Report
The paper:” Methane admixture protects liver mitochondria and improves graft function after static cold storage and reperfusion” aims to validate the possible protective role of CH4 during preservation of donor lever for transplantation. The paper is well written, the numerous and complete experiments are well organized and the results are convincing. All of this is preceded by a valuable scheme clearly showing the experimental plan, convenient to the reader to go through the text.
In general, data demonstrate the effective ability of CH4 to improve some functions, in particular on mitochondrial functions and extramitochondrial Ca2+ fluxes. With reference to text in lines 342-349, could the Authors propose an action mechanism by CH4. Could it be merely and mainly due to the claimed ability to enrich its concentration in lipid bilayers, or possible specific sensing mechanisms can be hypothesized?
Also, only very minor editorial points require attention, as detailed below, before the paper can be considered suitable for publication.
- from line 186, and from line 278, please remove italics.
- Figure 7, please adjust dimension.
- Line 435, if the case, please remove point 6.
